

# Bio-purification of drinking water by froth flotation
Ghanim Hassan[1], Robert G. J. Edyvean[2]
[1]Department of Water Resources Techniques, Middle Technical University, Baghdad, Iraq.
[2]Department of Chemical and Biological Engineering, The University of Sheffield, Sheffield, UK.
*Correspondence to:* Dr. Ghanim Hassan (dr.ghanim@mtu.edu.iq)
**Abstract.** The main technique for removing bacteria from water for various applications is chemical disinfection.
However, this method has many disadvantages such as producing disinfectant by-products (DBPs), biofilm formation
and either rendering the water unpotable (at high residual disinfection) or leaving a potential for lethal diseases such
as Cholera (if the residual disinfection is too low). Recently, a process was developed for continuous removal of
bacteria from water using the principle of froth flotation through compressed air only without any chemicals. This
work examines the extent to which chemical free froth flotation can purify drinking water.
The experiments were carried out using two flotation columns with different column lengths, each equipped with
ceramic air sparger. Raw water containing bacteria was fed into the column from the top. Air was pumped through
the water enough to produce a froth which separated the bacteria and, when removed, the bacterial content measured.
The results show that the bacterial concentration can be reduced by 55% of its original concentration under the optimal
experimental conditions so far found. This suggests that the technique can be used as a pre-purification step to
minimize the use of disinfectants; hence their byproducts, and to control biofilm growth.
**1. Introduction**
Froth flotation is a well-known solid-liquid separation technique using hydrophobicity as the driving force. Bacteria
and other water microorganisms tend to be hydrophobic and can be removed using such a separation method (Boyles
and Lincoln, 1958;Rubin et al., 1966;Bahr and Schugerl, 1992;Rios and Franca, 1997). However, the biggest limitation
against using froth flotation in bacterial removal from water is the difficulty of obtaining foam without using chemical
frothers. These chemicals may be toxic and /or deteriorate taste, odour and safety of the water. In addition to frothers
there are other chemicals that are used in froth flotation. These may be "collectors", "activators", "depressants" and
pH controllers. All or some of these agents may have a negative effect on water quality.
The second limitation against using froth flotation to purify drinking water is the particle size range. For mineral
particles the optimum size range for removal by froth flotation is 88-500 μm (Zech et al., 2012). At sizes greater than
this, the weight of particles is more than the bubble-particle adhering force while smaller particles can also
agglomerate forming bigger bulks which are also difficult to be kept adhering to a bubble (Zech et al., 2012;Otunniyi
et al., 2013). This particle size range is outside the range of microorganisms.
Using biocides in drinking water has many drawbacks. A particular problem is the formation of disinfectant by-
products (DBPs). Nowadays, there are more than 700 or more known DBPs (Brown et al., 2011;Gonsior et al.,
2014;Richardson and Postigo, 2015). Most have nothing known about their effects on public health; just 11 of them
are legalised in the United States. In addition, minimizing the negative effect of such chemicals using current
technology can produce yet more chemicals of known and unknown health problems (Ngwenya et al.,
2013;Richardson and Postigo, 2015).
The second drawback of using disinfection is the formation of a biofilm. The main driving forces behind biofilm
formation is a defensive strategy or to meet the metabolic needs of the bacteria (Chandra et al., 2001;Flemming,
2008;Simões et al., 2010;Kim et al., 2012). Thus, by definition, the formation of a biofilm may be a reaction against
an environmental threat to the microorganisms involved (Flemming, 2008;Simoes et al., 2010). Unfortunately, the
main threat to microorganisms is the use of biocides. Therefore, eliminating or decreasing the use of biocides will lead
to decreasing these two negative effects.
Nontoxic frothers can be used. Good results have been obtained using 1 mg/l milk casein as a frother for removing
bacteria from fishing port facilities (Suzuki et al., 2008).





Recently, froths have been produced by manipulating a compressed air stream in water without any chemicals. A
stable and well-built froth with a height of 27 cm was obtained (Hassan, 2015) and this was found to be able to remove
bacteria from the water column. Calculations predict that it is possible to obtain water with low or acceptable bacterial
levels starting from average river or reservoir bacterial concentrations (Hassan, 2015).
For smaller particles, such as microorganisms at around 10 μm or less, the optimum bubble size for separation is one
close to the size of the microorganism (Hanotu et al., 2012) or, indeed, to the size of other small particles such as
Carbon Nano Tubes (CNTs) (Lautenschlager et al., 2013).
The work presented here investigates whether the predicted results obtained from previous froth experiments can be
obtained practically for bacteria or not.

## 2    Materials and methods

### 2.1 Experimental set up

The experimental system consists of two Perspex (Poly (methyl methacrylate)) 20cm internal diameter columns of
one and two meters length. Air is supplied through a ceramic sparger 19 cm diameter and pore size of 50 microns
"from HP technical ceramics", fixed 10 cm above the column base. A water inlet is situated 15 cm underneath the
column top. A tank of 200 litres is attached the system for two main purposes, the first is as a reservoir for collecting
distilled water from the still, while the second is as a recycle tank when an experiment is run. Figures 1 and 2 illustrate
the experimental apparatus.   Note that the system in figure 2 is more complicated than the drawing because it is
designed to be used for other research as well.

### 2.2  iPad for colony counting

The iPad used for colony counting has the specifications shown in Table 1. The application (software) is (HGColony)
developed by (HyperGear Inc.) and can be downloaded through the Apple app store. The calibration and best operating
technique are described by (Hassan, 2015).

### 2.3  Peristaltic pump

A peristaltic pump from "Watson Marlow, Model - 505S", (range 2 - 220 rpm) was used for recycling and controlling
the flow of the water between the tank and the column. The water flow rate produced by this pump depends on the
pumping head. Since there are two column lengths in present work, a calibration curve for each column was
determined. For the two meters' column, the pump range (2 - 220 rpm) gave a flow rate range of (0.024 - 2.35 l/min)
while it was (0.029 - 3.15 l/min) for the one-meter column.  Both relations were linear and directly proportional. Any
flow rate could thus be calculated. For the one-meter column every 1 rpm equals to 0.0143 l/min while it is 0.0107
l/min for the two-meter column.

75                                    Table 1: iPad specifications

| iPad 4 with retina display 128 GB |
| --- |
| A6X chip |
| 1 GB Memory RAM |
| 5 MP rear camera and 1.8 MP front camera |
| iOS 6 Operating system |

### 2.4    Bacterial nutrient broth

Nutrient broth was prepared by mixing 15 g of nutrient broth powder from SIGMA-ALDRICH in 1 liter of distilled
water. When dissolved completely, it was autoclaved for sterilization. The sterile broth was inoculated with bacteria
and incubated for 24 hours at 37 C°. The bacteria used in this work are K-12 strain *Escherichia Coli* obtained from
Texas Red. The mother bacterial culture was kept deep frozen and used to prepare slants for further inoculation when
required.





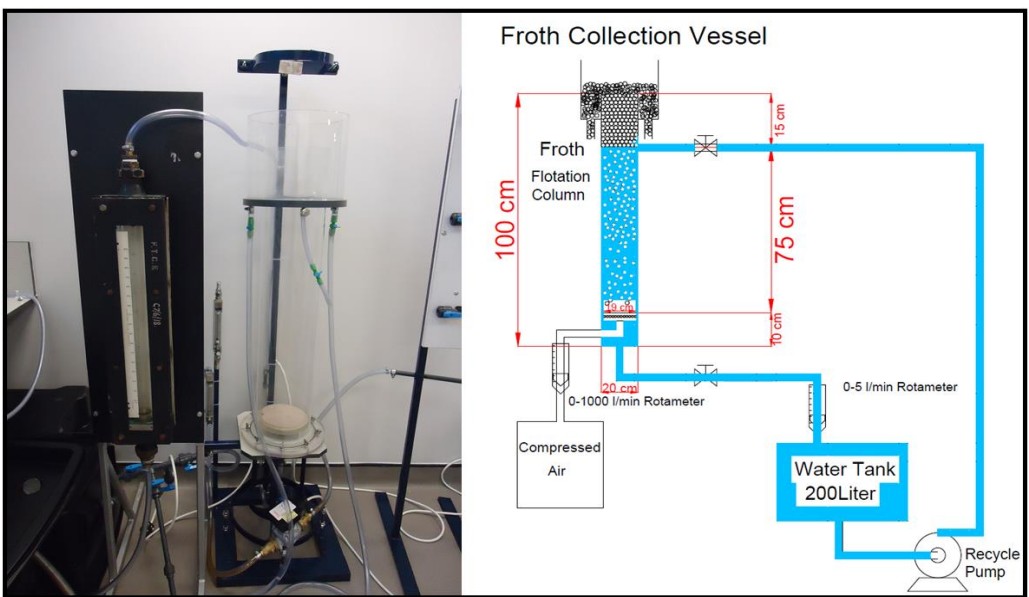

Figure 1: Experimental setup, 1m column

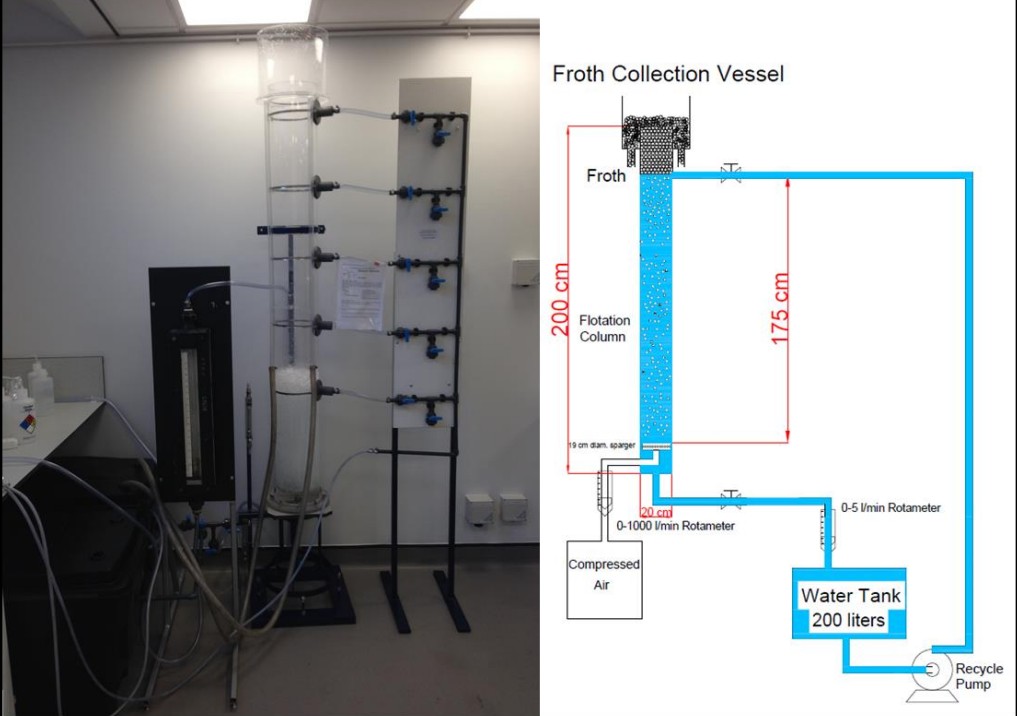

Figure 2: Experimental setup, 2m column

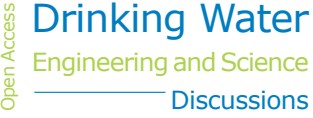


### 2.5 Agar plates

A mixture of nutrient broth and agar (15g + 15g) (SIGMA-ALDRICH) was suspended in a litre of distilled water then
boiled and mixed using magnetic stirrer. When dissolve completely, the nutrient agar was autoclaved (121 C˚ and
1kg/cm$^2$) then cooled to about 50 C˚ and poured in plastic petri dishes. When empty, the plates were incubated for 24
h at 37 C˚ to confirm accurate sterilization.
The plates were cultured aseptically with K-12 strain *Escherichia Coli* bacteria (Texas Red®) and incubated for 24
hours at 37 C˚.

### 2.6 Dead Bacteria preparation

Dead bacteria were obtained by treating a nutrient broth culture with a high dose of chlorine (1000-2000 ppm). One
liter of distilled water highly saturated with chlorine produced by adding trichloroisosyanuric acid was prepared and
mixed with the same amount of fully mature bacterial nutrient broth. The time needed to kill 99.9% of the bacteria is
less than one minute (EPA, 2009), the mixture is left for one hour to make sure of killing of all the bacteria. After that
the mixture was tested for two criteria. The first was to measure the mixture turbidity to make sure that there is no
reaction between the bacteria and the chlorine. Theoretically, if there is the same number of either live or dead bacteria
in solution, they should have the same turbidity. The second test was making sure all the bacteria in the mixture were
dead. One ml of the chlorine-bacterial nutrient broth mixture was added to a previously prepared and autoclaved
nutrient broth and incubated for 24 hours and 37 C$^o$. If there is growth, the death of bacteria was incomplete. If the
incubated nutrient broth showed no sign of growth the biocide process can be considered complete.
### 3  Experimental procedure

### 3.1 Preparing a water tank with known cfu/ml

1- Prepare one liter of inoculated nutrient broth as shown in (2.3.1).
2- Collect 100 liter of distilled water in the tank.
3- Add one ml of broth to the water tank, then mix and measure for colony count using a triplicate of Agar plates.
4- Add a second ml of nutrient broth to the tank and repeat step 3.
5- Continue adding nutrient broth till reaching the expected cfu/ml.
6- Next day, check the produced Agar plates to identify the quantity of inoculated broth to produce the desired
cfu/ml.

### a.  Measurement of percentage removal of bacteria

The following steps were followed to investigate the effect of studied variables (air flow rate and percentage of inlet
water removed by froth) on the percentage removal of bacteria.
1- Collect 100 liter of distilled water in the tank.
2- Add the amount of inoculated nutrient broth obtained in (2.3.1).
3- Take three samples for colony counting. These samples are for checking if the added amount of nutrient
broth gave the desired cfu/ml. If not the whole set should be repeated.
4- With an empty column, start air pumping at the rate of 10 l/min.
5- Start water pumping at 1 l/min.
6- Once water level plus froth reach the column top, open the downstream valve with a flow rate of 900 ml/min.
This will give a froth stream of 100 ml/min.
7- After 30 min, start taking samples for froth and downstream.
8- Stop the inlet flow. Then wait for the column for evacuating approximately quarter of its content.
9- Increase air pumping to 20 l/min. Then repeat steps 5 through 8, and so on for every next air flow rate.
10- Repeat steps 1 through 9 for every water downstream flow rates of (700, 600, 500, 400) ml/min.

### b.  Semi continuous flotation system

These experiments were designed to enhance froth performance and decrease the amount of water that is lost as froth.
The same technique used in 3.2 was followed but the inlet water stream was stopped when it reached the column top.

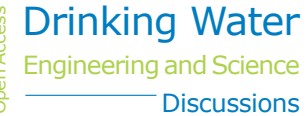

After that, the water level starts to decrease inside the column and froth should be built. When reaching an acceptable
froth height, restart the water flow. This will push the froth out of column again, and so on.  So, the following steps
should be followed:

    1- Collect 100 liter of distilled water in the tank.
    2- Add the amount of inoculated nutrient broth obtained in (2.3.1).
    3- Take three samples for colony counting. These samples are for checking if the added amount of nutrient broth gave the desired cfu/ml. If not the whole set should be repeated.
    4- With an empty column, start air pumping at the rate of 10 l/min.
    5- Start water pumping at 1 l/min.
    6- Once water level reaches the column top, open the downstream valve with a flow rate of 1 l/min.
    7- Stop upstream water inlet.
    8- Water level in column should start to decrease and the froth start to build.
    9- Once the froth reaches its steady state height or moves away from column top edge, start upstream water inlet again at 1 l/min.
    10- As the water level increases inside the column, it displaces the froth to froth collector.
    11- When the froth disappears as a result of water level rising inside the column, stop water inlet again and so on.
    12- Take samples for colony count every time the inlet upstream water is stopped.

## 4 Results

Measurements of cfu/ml were taken for the inlet, bottom and froth streams. The froth reading was taken for guidance.
The "purification force" depends on the difference between inlet and bottom streams:

$$\text{Percentage removal of bacteria} = \frac{Inlet\left(\frac{cfu}{ml}\right) - downstream\left(\frac{cfu}{ml}\right)}{Inlet\left(\frac{cfu}{ml}\right)} * 100$$

### 4.1 Effect of air flow rate and percentage of inlet water removed by froth on the percentage removal of bacteria.

In these experiments, two operating variables were investigated to determine their effect on the purification force of
froth flotation, the air flow rate and the percentage of inlet water removed by froth. Figures 3 and 4 show the results
for the 1 and 2 meter columns respectively.

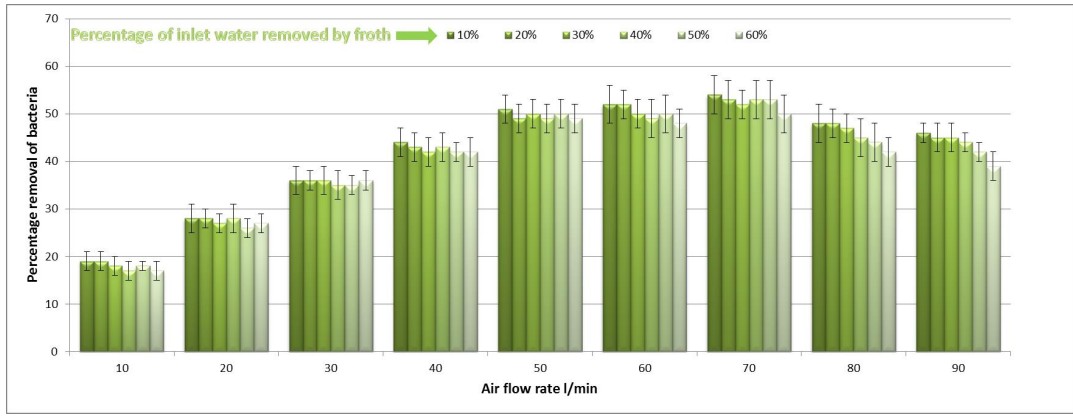

Figure 3: Effect of air flow rate and percentage of inlet water removed by froth on the percentage removal of bacteria (stage 1, one meter column length)

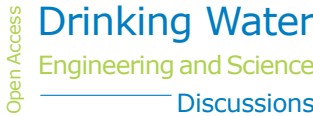

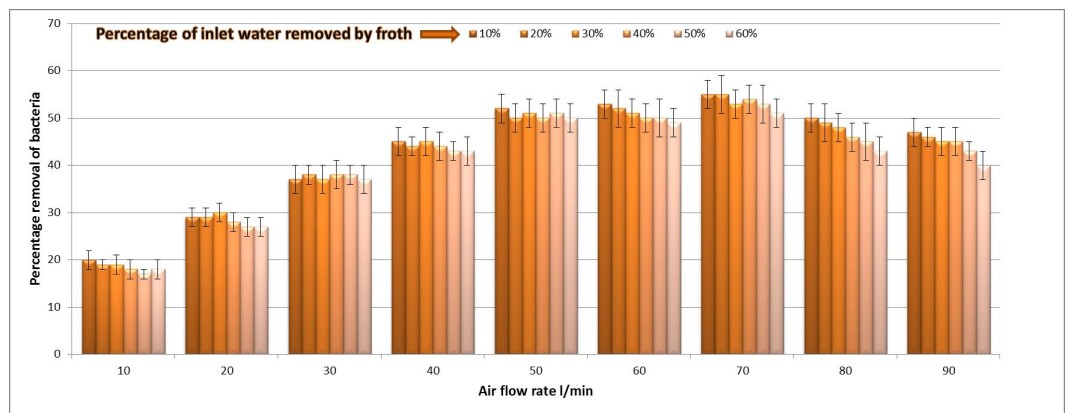

Figure 4: Effect of air flow rate and percentage of inlet water removed by froth on the percentage removal of
bacteria (stage 2, two meters column length)
**4.2 Semi continuous flotation system**
Figure 5 represents the results obtained when running the experimental procedure, described in 3.3, for a semi
continuous flow system. The studied variables were the air flow rate for both stages. The percentage of removal of
bacteria was determined.

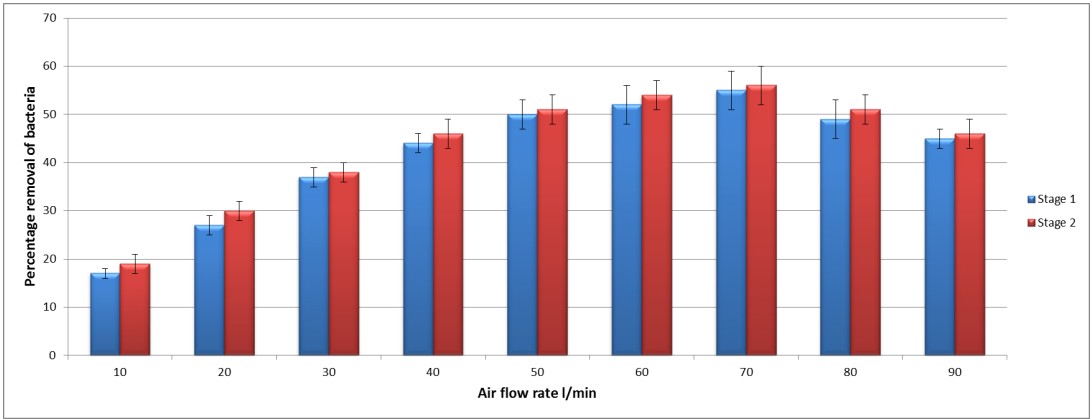


Figure 5: Effect of air flow rate on the percentage removal of bacteria in semi continuous system
**4.3 Response of dead bacteria to froth flotation**
Five optimum points from chapter four were selected to repeat with dead bacteria. The fixed operating parameters
were, air flow rate of 130 L min$^{-1}$ , operating time of 30 min, and the water level inside the column was 150 cm. The
results are summarized in Figure 6. It has been shown that the average difference between chlorination and
dechlorination rates was 5-10 % under the same circumstances.



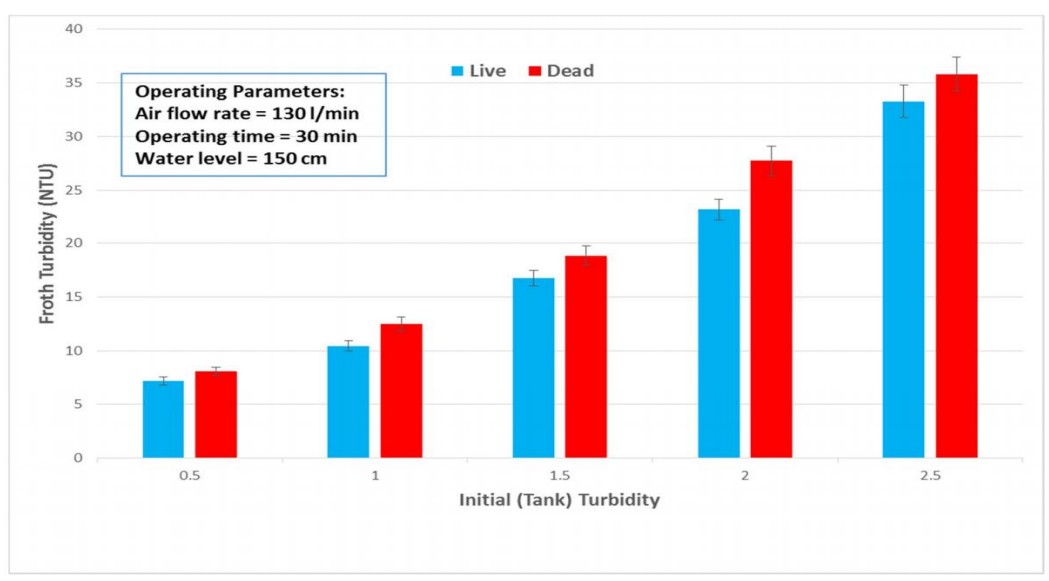


184             Figure 6: Comparison between using froth flotation with dead and live bacteria.

**5     Discussion**
**5.1 Continuous purification process**
The main aim of this work is to investigate whether it is possible to reach a suitable purification of water from
microorganisms using a chemical free froth flotation. The driving force that leads this process is hydrophobicity and
most of the microorganisms in water are hydrophobic and should be suitable for removal by such a technique (Boyles
and Lincoln, 1958;Rubin et al., 1966;Bahr and Schugerl, 1992;Rios and Franca, 1997).
Three variables were optimized in this study, air flow rate, ratio of water removed by froth, and water column length.
For the air flow rate the trend shows a decrease in downstream cfu/ml (increasing bacterial removal) with increasing
air flow in the range from 10-50 l/min. This is because the increase of air pumping will increase bubbles which lead
to the probability of more bubble-bacterial attachment. At higher air flow, in the range of 50-70 l/min, there was a
near plateau state. This is due to the appearance of turbulence in the column and the remixing bubbles in froth back
into the water bulk. Furthermore, water starts to move up and down into the froth. These "Waves" destroy the froth
itself. Some water drops are not returning back to column completely but are transferred to froth collection vessel
which ruins the froth concentration, hence the whole separation process. This phenomenon increases as the air flow
rate increases.
The turbulence is not taking place at the top of the column only but it is a bubble-water phenomenon and occurs along
the whole column length. It affects the separation process directly and negatively because the process depends mainly
on consolidating bubble-bacteria attachment and avoiding detachment. With high turbulence the detachment force
will increase and this decreases the process efficiency.
The inlet upstream is divided into two outlet streams. The first is the purified downstream water while the second is
water leaving the column top as froth. The second studied variable is the effect of the ratio of the inlet upstream that
is discharged as froth on the downstream cfu/ml. If this ratio increases it will have two counteractive effects; that is,
more discharged water but with less cfu/ml. The "more discharged water" should enhance the purification while the
"less cfu/ml" lowers process efficiency. The sum of these two effects was tested and found to be nearly the same but
opposite.



The third variable is the height of the column. It would be expected that the longer the column the greater the bacterial removal efficiency as there is more chance of bubble-bacterial attachment. The results endorse this hypothesis in direction but not in amount. Doubling the column length would be expected to double the froth bacterial concentration but in practice the longer column only slightly enhances the removal efficiency. It seems that every bubble has a certain holding capacity that cannot be exceeded. In previous work, it is found that a significant difference in froth bacterial concentration at different water column heights as the water column was increased in height (Hassan, 2015). However, there was no continuous discharge of the froth. Therefore, the bacterial concentration in the froth was cumulative. In other words, if a certain bubble rises along the water column it will start attaching bacteria to some extent. Then it will rise up till reaching the froth. Once there, this bubble continues climbing up through the froth with the assistance of other bubbles which are rising up under it until it arrives at the froth top where it will burst leaving its bacteria on the froth. With no froth discharge from the column top the bacterial concentration increases within the froth. However, in the current chapter, there is a continuous discharge of froth, so there is no time for accumulating bacteria in the froth.

In a previous work in Japan a separation of 80% was obtained in a 14l column which indicates that smaller columns can do the desired job as well as, or better than larger columns (volumes in present study was 31.5 l and 63 l for columns 1 and 2 respectively) (Suzuki et al., 2008). However, they were using froth flotation to remove bacteria from sea water in fish farms and not only added casein protein as frother, but seawater also contains self frothers like fish's mucus and salts. This may explain the better bacterial removal efficiency they found.

While not without cost, aeration is a conventional process used in the drinking water industry for various proposes such as the removal of volatile organic chemicals (VOCs), gases, and oxidizing dissolved metals such as Iron (Albin and Holdren, 1985;Baylar et al., 2010). Depending on the purpose of aeration, in many industrial applications the air to water ratio used is close to that used in this study (Marjani et al., 2009;Sales-Ortells and Medema, 2012) This suggests the economical limitations of using compressed air are already accepted in the water treatment industry. Indeed, some countries accept the high cost of Ozonation to avoid the health issues of other biocides (EPA, 1999). Therefore, aeration basins could be modified in order to add the removal of microorganisms to the known duties of aeration. This method is clean and does not attack bacteria aggressively and drive them to produce biofilms which can harbor pathogens. Also, it decreases the need for biocides, hence lowering their direct and indirect drawbacks such as disinfectant by-products. Finally, it is not a complicated technology and is easy to install and operate. The bacterial removal rate in this work reached 55% expressed as a percentage difference in input and purified streams. Using it as a solo technique for removing microorganisms is a controversial issue. Most of rivers worldwide vary in their cfu/ml for total coliforms. Three rivers in India have been analyzed and give values between 100 - 120 cfu/ml (Rajiv et al., 2012). The Foma River in Nigeria gives counts ranging from 2700 to 12300 cfu/ml (Agbabiaka and Oyeyiola, 2012). River water sources in rural Venda communities in South Africa gave a minimum and maximum of 600 cfu/ml and 37000 cfu/ml respectively (Obi et al., 2003) In Myanmar samples from deep wells and dams in two urban areas; namely, Nay Pyi Taw and Yangon gave 3 to14 cfu/ml (Sakai et al., 2013). For drinking water, the colony count does not necessarily equate to the health risk because humans have immunity to many bacterial species. However, for example, German drinking water regulations consider 100 cfu/ml as an acceptable limit for tap water (Bartram et al., 2003). Also, an upper limit is reported to be 500 cfu/ml, though the range of 100 to 500 CFU/ml is still "questionable" (Edstrom, 2003). Therefore; industrially, the acceptance of such methodology as non-chemical froth flotation depends on the source water and desired water quality.

## 5.1 Semi continuous purification process

The froth was found to be stable up to a flow rate of 130 l/min (Hassan, 2015) but in this work it starts to collapse after 70 l/min., this because the froth is working at the column top (upper edge). Here the froth structure losses the wall support and collapses into the froth collector. Also, continuous operation drives a lot of water to exit from the column top with the froth bubbles.

Some operational modifications can be suggested to avoid this problem. When the upstream inlet is shut off, the water level inside the column goes down. When this happens the column wall helps the froth to build up again. This froth will continue collecting bacteria at the same rate of continuous process but accumulatively. If the upstream inlet is only restarted when the froth reaches its maximum sustainable height, the water level will increase again and push the froth up to be discharged with minimal additional water. Once the froth is pushed out the top of the column completely the inlet can be stopped again and the most efficient cycle repeated.



The effect of this optimization on bacterial removal efficiency is not great. However, it is very useful for minimizing
the ratio of disposed water with the froth. Therefore, it is recommended to use an optimized semi continuous process
when water is valuable.

### 6.5.3 Dead bacteria separation

The aim of this experiment is to study the separation of dead bacteria by froth flotation. The results show that the dead
bacteria can be separated to a greater extent that of live bacteria by a factor of 5-10%. Live bacteria do have some
independent motility, depending on the species, and have active attachment/detachment mechanisms. These results
could indicate that a small proportion of live bacteria may be able to avoid attachment to bubbles. Using froth flotation
to purify water from dead bacteria can decrease the amount of any additional disinfectant required and decrease
disinfectant by-products and increase water biostability. Therefore, the use of froth floatation has the double advantage
of removing both live and dead cells (and, by implication, removing other particulate contaminants) (Griebe and
Flemming, 1998; Castro and Neves, 2003). The destiny of dead bacteria has received little attention in the drinking
water industry. In conventional chlorination most of bacteria will die and the water is considered safe, but this is not
the final word. The role of dead bacteria in the drinking water system requires further research.

## 6   Conclusions

Froth flotation is a promising technique in water industry. This study shows that some 55% of bacterial cfu/ml can be
reduced by froth flotation without chemicals. Semi continuous flow gave slightly less purification efficiency but with
much less water discharged with the bacterially laden froth.

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
