# Peer review of "Bio-purification of drinking water by froth flotation"

_Drinking Water Engineering and Science, 2018_

## Referee Comment (RC1) · Anonymous Referee #1 · 8 Jan 2019

The authors studied the removal of bacteria from water using the principle of froth flotation through compressed air only without any chemicals. This study shows that some 55% of bacterial cfu/ml can be reduced by froth flotation without chemicals. The manuscript is well organized.

Specific comments are below:

Page 2, line 75: the table 1 is not necessary to show the details of the iPad.

Abstract: These sentences should be in introduction part other than in abstract: "The main technique for removing bacteria from water for various applications is chemical disinfection. However, this method has many disadvantages such as producing disinfectant by-products (DBPs), biofilm formation and either rendering the water unpotable

(at high residual disinfection) or leaving a potential for lethal diseases such as Cholera (if the residual disinfection is too low). " Or, is should be in a shorten way. And, I suggest to use only one paragraph in the abstract.

Page 5, figure 3: The green color words on the top of the figure are not clear, redraw it: "Percentage of inlet water removed by froth"

Page 7, line 184ïijŽVertical coordinate is needed in the figure. And the figure is not very clear, redraw the figure, and the color used in the figure is ambiguous.

---

## Referee Comment (RC2) · Anonymous Referee #2 · 30 Jan 2019

In this paper froth flotation, without the use of chemicals, is studied to remove bacteria from surface water, with the idea to diminish the doses of e.g. chlorine to disinfect the water. The study is creative, since froth flotation is not being used in drinking water production. However, unfortunately, the obtained results (50% removal) are not sufficient for implementation in practice, since in drinking water production frequently log 3 (99.9%) or more removal are required. The paper misses reflection on this issue. In addition, the paper builds on two other submitted papers. It is recommended to merge the three paper into this one, to avoid overlap and missing information (such as froth height in the columns), when reading the paper. General comments: - A clear explanation of the difference between froth flotation and DAF should be included in the introduction - The results chapters lacks explanation of the figures (given in discussion

chapter) - It is recommended to merge the results and discussion chapter to avoid the above - A more extensive discussion on the results in the light of literature and competing technologies should be included - It is not clear if there is a reference column (without froth flotation) installed. The long retention times in the columns can also give "spontaneous" decay of the bacteria. - It is not clear what the section on "dead bacteria" adds to the paper. So the suggestion is to delete this part. Specific comments: - Line 7, it is questionable if chlorine enhances biofilm formation (nowadays chlorine is also used to control biofilms in drinking water distribution) - Line 12 (and onwards), "length" = "height" - Line 14, content was measured - Line 16-17, questionable, see above. - Line 31, the word "biocide" is not used in drinking water treatment. Better use "disinfection" of "chlorination". - Line 33-34, rephrase or delete (unclear and not of importance) - Line 35-36, rephrase (unclear what is meant) - Line 40-42, see remark above - Line 47-48, see remark above - Line 52, a clear knowledge gap should be presented (may be also in combination with the other two papers) - Line 57, insert "respectively" after two meters height. "... sparger of 19 cm..." - Line 59, "... attached to the..." - Line 61-62, delete sentence. Not relevant here. - Line 64 and 75, delete Table 1. Not relevant information. - Line 77, explain why a nutrient broth is needed for the experiments - Line 79, explain that 24 hours incubation is sufficient in order to... - Line 82-84, avoid duplication of figures (almost similar) - Line 87-93, explain why this procedure was followed and if this is also done in other studies in this way. - Line 94-105, delete (see comment above) - Line 106 (and onwards), use passive tense: "was prepared", "was collected" etc. - Line 107 (and onwards), check section numbering. - Line 113 and 114, explain what is meant by "expected" and "desired"? - Line 121, "(2.3.1)" is not found in the paper. - Line 128 (and onwards), "downstream"= "effluent" - Line 129-131, explain why exactly this procedure is followed. - Line 130, mention the "next air flow rates". - Line 131, does it mean that with lower effluent flows there are higher froth flows? - Line 135, "3.2" cannot be found in the paper. - Line 137, "was restarted" - Line 138, "should be" = "were" - Line 140, "2.3.1"? - Line 145 and 148, "reaches" = "reached" - Line 157, "the difference of the concentrations between the

inlet and the effluent flows" - Line 166 and 170, what is meant by "stage 1" and "stage 2"? - Line 178-184, delete section - Line 187-190, this paragraph is more for the (end of) the introduction - Line 192, "downstream"= "effluent concentration" - Line 195, how the turbulence was measured/calculated? And how the "remixing of the bubbles in the froth" was observed? - Line 196, How the "waves" were observed? - Line 200-203, give reference on this explanation. - Line 207, "with less cfu/ml in the effluent" - Line 208-209, not clear how this was determined. - Line 210-212, it is not clear if the expected removal of bacteria was through the froth layer or in the entire column. It is also not clear what the froth layer thickness was (and if these were different for the two columns) - Line 213, "enhances" = "enhanced" - Line 221, "in the current chapter" (is this copied from the original thesis)? - Line 227 is floc formation and attachment of bacteria in natural water (with salt) also not a relevant mechanism of enhanced bacterial removal? - Line 228-249 not relevant here, so delete - Line 250, check numbering of sections - Line 251, "starts" is "started", in the other papers higher air flows were reported.. - Line 252, "loses itself from" - Line 253, in continuous operation more water is wasted, but also more water is produced, so report the efficieny (in %). - Line 256, since the column wall is apparantly important for froth build up, then reflect on large-scale application. - Line 262-263, this recommendation could not be made based on the presented work - Line 264-265, delete section - Line 276-278, this conclusion could not be made based on the presented work

---

## Author Comment (AC1) · 13 Feb 2019

All the comments are taken into account.

[Figure]
Interactive
comment

**Bio-purification of drinking water by froth flotation**

Dr. Ghanim Hassan*, Department of Water Resources Techniques, Middle Technical University,

Baghdad, Iraq, dr.ghanim@mtu.edu.iq

Dr. Robert G. J. Edyvean, Department of Chemical and Biological Engineering, The University of Sheffield, Sheffield, UK,  r.edyvean@sheffield.ac.uk.

Key words: Froth flotation, Bacteria bio-purification, Drinking water.

         **Abstract**

Recently, a process was developed for continuous removal of bacteria from water using the principle of froth flotation through compressed air only without any chemicals (Hassan, 2015).

This work examines the extent to which chemical free froth flotation can purify drinking water.

The experiments were carried out using two flotation columns with different column heights, each equipped with ceramic air sparger. Raw water containing bacteria was fed into the column from the top. Air was pumped through the water enough to produce a froth which separated the bacteria and, when removed, the bacterial content measured.

The results show that the bacterial concentration can be reduced by 55% of its original concentration under the optimal experimental conditions so far found. This suggests that the technique can be used as a pre-purification step to minimize the use of disinfectants; hence their byproducts, and to control biofilm growth.

•     Correspondence Author: Dr. Ghanim Hassan, dr.ghanim @mtu.edu.iq, 00964-
7704335364.

**Fig. 1.**

---

## Author Comment (AC2) · 13 Feb 2019

Indeed 50% removal is not sufficient but remember that Wright brothers started the flying age by a 12 seconds flight. However, I am now starting a research line to raise up this percentage. Merging the three papers means I should submit my Ph.D. thesis as one paper, I think this is not possible. I think there is no need for reference column since the calculations embed both the inlet and downstream streams. I think a few hours does not be considered as "long retention time". The dead bacteria are very important since they result in turbidity as the same as life and can interfere with the results. Furthermore, removing dead bacteria may remove a potential future nutrient and a source of DBPs. Indeed, Chlorine stimulates biofilm formation in low concentrations as it is referenced clearly in the paper. Removing biofilm by high Chlorine doses is a common and famous technique. Nutrient broth is needed to produce a high bacterial content solution which in turn be diluted to adjust the desired CFU/ml. It is familiar and common practice that in order to produce a full mature bacterial broth, you should incubate for 24 hours. Waves were observed by the vision

[Figure]

**Bio-purification of drinking water by froth flotation**

Dr. Ghanim Hassan\*, Department of Water Resources Techniques, Middle Technical University,

Baghdad, Iraq, dr.ghanim@mtu.edu.iq

Dr. Robert G. J. Edyvean, Department of Chemical and Biological Engineering, The University of Sheffield, Sheffield, UK, r.edyvean@sheffield.ac.uk.

Key words: Froth flotation, Bacteria bio-purification, Drinking water.

**Abstract**

Recently, a process was developed for continuous removal of bacteria from water using the principle of froth flotation through compressed air only without any chemicals (Hassan, 2015).

This work examines the extent to which chemical free froth flotation can purify drinking water.

The experiments were carried out using two flotation columns with different column heights, each equipped with ceramic air sparger. Raw water containing bacteria was fed into the column from the top. Air was pumped through the water enough to produce a froth which separated the bacteria and, when removed, the bacterial content measured.

The results show that the bacterial concentration can be reduced by 55% of its original concentration under the optimal experimental conditions so far found. This suggests that the technique can be used as a pre-purification step to minimize the use of disinfectants; hence their byproducts, and to control biofilm growth.

• Correspondence Author: Dr. Ghanim Hassan, dr.ghanim @mtu.edu.iq, 00964-
7704335364.

[Figure]

**Fig. 1.**